# From Technocracy to Democracy: Ways to Promote Democratic Engagement for Just Climate Change Adaptation and Resilience Building

David Olsson

Political Science, Department of Political, Historical, Religious and Cultural Studies, Karlstad University, 651 88 Karlstad, Sweden; david.olsson@kau.se

**Abstract:** Climate change and the policy responses to it have implications in terms of (in)justice. Research in fields such as political ecology and environmental justice emphasizes the importance of policy-making addressing and responding to climate injustices. It, moreover, stresses that democratic engagement is imperative, since no universal agreement on the meaning of "justice" exists. Democratic engagement on climate (in)justice is, however, hampered by the predominance of technocratic policy frames. Considering this, knowledge of ways to promote democratic engagement is called for. This study develops such knowledge related to policy-making for climate change adaptation and resilience at the local level, in developed country contexts. Specifically, it draws on the "what's the problem represented to be?" approach to conceptualize different styles of democratic engagement and examine the possibilities and limitations of each. From the data, comprised of previous research, representations of three styles of democratic engagement are identified and analyzed: (1) closure-oriented engagement centered on changing behaviors, (2) closure-oriented engagement centered on changing the systemic production of unjust practices, and (3) disruptive engagement centered on changing the systemic production of unjust practices. The contributions of this study are relevant to researchers, policymakers, activists and others interested in how to promote a democratization of climate policy-making.

**Keywords:** climate change adaptation; deliberative democracy; civic republicanism; agonistic democracy; social practice theories; behavior-change theories; WPR; governmentality; transformation; just sustainabilities



## 1. Introduction

The fact that human activities cause climatic changes that undermine the resilience of societies and ecosystems has incited calls for policies that alter such unsustainable activities [1,2]. A circumstance that complicates policy making is, however, that both the impacts of and policy responses to climate change have implications in terms of (in)justice [3–8]. Accordingly, scholars in research fields such as environmental justice [9,10] and political ecology [11,12] argue that policy responses to climate change should promote just transformations of human activities.

The meanings of justice and just transformations are, however, contested, as indicated by rivaling theories and notions of justice (e.g., [13–16]), and different proposals of how environmental and climate justice should be understood (e.g., [6,17–22]). Reflecting this disagreement, scholars stress the importance of democratic engagement with policies on climate change [23–25].

Democratic engagement on climate policies is, nonetheless, often hampered by the prevalence of technocratic policy approaches (e.g., [26–30]). These "apolitical" approaches set the focus on managerial and technical issues (e.g., [5,23,27,31,32]). As a consequence, climate (in)justice is often an absent topic in climate policy-making, especially political disputes over injustices and the actions (re)producing these [5,28,29,33].

The predominance of technocratic approaches to climate change has led several studies to stress the importance of a shift to democratic engagement with issues of (in)justice (e.g., [5,9,23]). Moreover, since climate change is induced by human activities, studies on climate injustices typically emphasize the need to alter unsustainable actions, such as those pertaining to luxury consumption [6,34–37]. Taken together, these and other studies provide different representations of how unsustainable actions can be altered democratically in ways promoting social change for climate justice. This body of research could inform and inspire policy makers, planners, citizens and other actors to democratize the policy processes of climate change, especially actors and movements that want to promote a shift beyond technocratic approaches, many of which are based on the market logic [7,26,33].

The ideas of democratic engagement in these studies can, however, be hard to navigate and assess, especially since there are rivaling theories of both democracy (see [38–41]) and social change (see [42–44]) that are plausibly reflected in this research. If so, these studies would be underpinned by different "conducts of conduct" (cf. [45]), which I label "styles of democratic engagement". Drawing on insights from the governmentality literature [45–48], particularly the "what's the problem represented to be?" (WPR) approach [49], the conduct of conduct of a particular style of democratic engagement can be understood in terms of a specific style of problematizations. Importantly, these problematizations produce particular possibilities and limitations. Knowledge of these would be useful to both researchers and actors outside of academia with an interest in promoting democratic engagement.

Democratic engagement for climate justice is arguably important at different levels of policy-making on climate change mitigation and adaptation across the globe. This study is, however, delimited to an analysis of research on climate change adaptation and resilience policies at the local level in developed countries. (Although some research describes resilience as a form of climate change adaptation that is characterized by a technocratic approach to climate change [9,50], other studies highlight that there are different modes of resilience, including those oriented toward democracy and social transformation [51,52]. This study employs the latter conception of resilience.) I, moreover, employ a theory-driven analytical strategy based on rivaling theories of democracy and social change. The reason for the latter is that it, in addition to further delimiting the focus of this study, enables a more in-depth discussion of the findings in relation to scholarly critique of these theories of democracy and social change.

Considering the above, the purpose of this study is to conceptualize four styles of democratic engagement and, subsequently, examine the problematizations that form part of each of these, as they are represented in research on climate just adaptation and resilience at the local level in developed countries. This examination is also centered on the possibilities and limitations that the problematizations of each style of democratic engagement produce. To fulfill the purpose, the analysis and discussion is guided by the following questions:

(1) What styles of democratic engagement emerge from the combination of two political strategies and two strategies for social change?

(2) What problematizations of these styles of democratic engagement are represented in the examined research?

(3) What possibilities and limitations are produced through the problematizations of each style of democratic engagement?

The outline of this paper is as follows: First, the theory-driven analytical approach is outlined. Then the first question is answered through a conceptualization of four styles of democratic engagement, comprised of two theoretically grounded dimensions: political strategies and strategies for social change. Subsequently, the methods for retrieving the analyzed research are outlined. This is followed by a presentation of the problematizations of the styles of democratic engagement represented in the studies, which answers the second question. Next, the answer to the third question is provided through an analysis of the possibilities and limitations of the problematizations of each style. In answering this question, the results are also discussed in relation to critique of the theories of democracy

and social change that each style reflects. Finally, the article is concluded with a brief discussion of the implications of the findings for further research as well as for policy-makers, planners, activists and other actors outside of academia.

## 2. Analytical Approach

This study employs a theory-driven analytical strategy that draws on concepts and analytical foci found in the WPR approach [49,53,54]. Namely, I first constructed four styles of democratic engagement from different problematizations and assumptions represented in rivaling theories of democracy and social change, as detailed in the next section. Subsequently, I identified problematizations in studies on adaptation and resilience reflecting those of the theories underpinning the styles of democratic engagement.

Importantly, the WPR approach defines problematizations as those implied through particular prescriptions and guides to practice, not as explicit descriptions of problems. The assumptions in focus are those underpinning these problematizations [49,53,54]. Such problematizations and underlying assumptions represent different "conducts of conduct". Namely, a particular pattern or "style of problematization" represented across several policies—or, as in this article, studies—signifies a "mentality" as in the concept of "governmentality" ([49], p. 6). However, whereas the WPR is a policy-analytical approach, I use it to identify "mentalities" or "conducts of conduct" represented in theories of democracy and social change as well as in studies on climate change adaptation and resilience.

Finally, it should be stressed that this study provides a redescription of the analyzed research. The theories utilized to construct the four styles of democratic engagement are not mentioned in most of the studies that constitute the data. As argued in the analysis, the problematizations and assumptions of these theories are nevertheless reflected in them.

## 3. Four Styles of Democratic Engagement

The four styles of democratic engagement were constructed from a matrix based on two dimensions: political strategies and strategies for social change (see Table 1). The political strategies include closure-oriented and disruptive strategies. The social-change strategies encompass behavioral-oriented and practice-oriented strategies.

**Table 1.** Four Styles of Democratic Engagement.

| | | Political Strategies | |
| | | Closure-Oriented Strategies | Disruptive Strategies |
|---|---|---|---|
| **Strategies for social change** | **Behavior-oriented strategies** | Closure-oriented engagement centered on behaviors | Disruptive engagement centered on behaviors |
| | **Practice-oriented strategies** | Closure-oriented engagement centered on practices | Disruptive engagement centered on practices |

The closure-oriented strategies are those focused on ways to reach decisions on policies. Naturally, strategies to reach closure are emphasized in several democratic theories. This study is delimited to closure-oriented strategies that ground democratic politics on ethics, particularly civic republican (e.g., [38,55]) and deliberative democratic theories (e.g., [56,57]). Hence, democratic theories that, for instance, base politics on the notion the self-interested individual of homo economicus and the market logic, as in public choice theories (see [58–60]), are excluded.

Civic republican and deliberative democratic theories both offer ethically based strategies for democratic closure. Civic republican theories are underpinned by the assumption that a set of shared substantive virtues are essential in order for democratic politics to promote public interests and provisional "common goods". Such political virtues should be negotiated, cultivated and nurtured through political deliberation and education. The formation of political virtues is thus presumed to be a premise for policy closure on provisional public goods—closure that does not require consensus [38,55].

Deliberative democratic theories reject the notion of substantive virtues (see [40]) but are nevertheless grounded on the notion that political processes should be underpinned by a procedural ethics. An example of this is Habermas's discourse ethics for communicative action. It underscores that all effected parties should be able to criticize and present validity claims as well as empathize with the other participants' validity claims. The closure-orientation of his procedural ethics is based on the aspiration to reach consensus around the "better" argument [56,57].

Although there are significant differences between civic republican and deliberative democratic theories, they both problematize (the lack of) democratic closure on common goods in ethical terms, either as a lack of substantive or procedural ethics. In relation to climate justice, this problematization thus constitutes a conduct of conduct centered on promoting ethical citizens and/or processes to reach closure on just climate policies.

The disruptive strategies instead center on ways to support peaceful conflict and contestation between different emergent "us" and "them". Importantly, these strategies differ from, for instance, the communicative action of Habermasian discourse ethics in that their target is to disrupt the formations of power that constitute the possibilities and limitations of "the rational" [61,62]. That is, these strategies focus on disrupting naturalized ways of thinking and doing things, thereby creating possibilities for conceiving things previously inconceivable.

Disruptive strategies are represented in different agonistic theories (see [63]). For instance, Mouffe's agonistic pluralism [39,64,65] is based on the notion that peaceful contestation between political adversaries, which enables hegemonic formations of power to be challenged and dislocated, is imperative. This notion is underpinned by the ontological assumption that there is no universal foundation on which democratic politics could be based. Rather politics is by necessity grounded on particular and contingent hegemonic projects, which always produce exclusions that subjugate and oppress some ideas, identities etc. This ontological assumption is combined with the normative stance that hegemonic discourses and practices continuously should be opened to contestation since they inevitably produce particular oppressions and harms. Other examples of disruptive strategies are expressed by scholars such as Rancière [66] and Connolly [67]. Although there are significant differences between these scholars, they share a focus on ways to challenge and disrupt dominant formations of power to open up new possibilities, which here is the defining characteristic of the disruptive strategies. Hence, these strategies share a problematization of exclusions produced through naturalized formations of power. Related to climate justice, this problematization constitutes a conduct of conduct focused on promoting contestation and disruptive citizen actions, hence the label "disruptive strategies".

I now turn to the strategies for social change. I base these on the vibrant social science debate on behavior-oriented theories, so-called Attitudes, Behavior, Choice (ABC) theories, versus social practice theories (henceforth, practice theories). These two groups of

theories make diametrically different proposals as to how unsustainable actions should be understood and changed (see [42–44,68]).

The ABC theories have their roots in the disciplines of (social-)psychology and economics whereas practice theories have emerged from subjects such as sociology, science and technology studies, complexity science and history [43,44]. The prescriptions, and implicit problematizations, represented by each of these two groups of theories differ significantly. Accordingly, each constitutes a different conduct of conduct.

The behavior-oriented strategies are conceptualized based on the ABC theories. As indicated by the label Attitudes, Behavior, Choice (ABC), these theories share a focus on external drivers supporting and motivating individuals to choose sustainable behavioral options. Such support and motivation are understood in terms of overcoming contextual and attitudinal barriers for choosing sustainable options as well as promoting drivers triggering individual choices of sustainable behaviors [44]. One assumption is thus that social change for sustainability, such as climate justice, comes about through knowledge, incentives etc. that encourage and enable individuals to choose differently. Accordingly, individuals are assumed to be the primary agents of change. The focus on external drivers, to enable them to become such agents, also means that individuals are presumed to be relatively autonomous in relation to the needs, desires, identities and aspirations and other attachments to their everyday behaviors [44,69–71].

The proposals of these theories to support and motivate individuals to choose to abandon certain behaviors in favor of alternatives imply that the problematization of unsustainable actions is centered on individual choices of behavior. This signifies a conduct of conduct that, as Shove puts it, "[ . . . ] *locates citizens as consumers and decision makers and which positions governments and other institutions as enablers*" ([44], p. 1280).

The practice-oriented strategies are conceptualized based on practice theories. These theories do, by comparison, have a systemic orientation centered on the internal dynamics of particular entanglements—so-called socio-technical arrangements—from which practices emerge. These include interactions between specific norms and institutions, identities, cultural values, infrastructures, modes of transportation, various technical artifacts, and so on. Hence, the attention is directed to ways in which particular systemic dynamics that produce unsustainable practices—along with the needs, desires, aspirations and other attachments that form part of them—can be unmade and replaced. This emphasis on changing systemic dynamics is quite different from the ABC theories' focus to promote sustainable individual choices of behavior that "go against the stream". A reason for this is that practice theories are underpinned by the assumption that people's attachments to, and embeddedness in, day-to-day practices are relatively strong and very difficult to escape for individuals. Accordingly, practice theories set the focus on ways to unmake the internal dynamics of socio-technical arrangements from which unsustainable practices emerge and on ways to replace these with those that produce sustainable practices [44,71–73].

The proposals of practice theories imply that unsustainable actions are problematized as being the result of a lack of engagement to unmake and replace the internal dynamics of socio-technical arrangements that reproduce unsustainable practices. This signifies a conduct of conduct in which citizens are understood as carriers of practices, entangled with particular dynamics of socio-technical arrangements. Social change is, moreover, made possible through actions that change these dynamics, rather than by individual and collective actions to support and motivate sustainable individual choices, as in the ABC.

If the problematizations and assumptions of each group of political strategies are combined with those of the strategies for social change, four styles of democratic engagement emerge, each with its particular conduct of conduct. An overview of these is provided in Table 1.

## 4. Materials and Methods

This section outlines the methods used for retrieving the analyzed studies on climate change adaptation and resilience as well as for documenting their content. Importantly,

the research design entails that the focus is on identifying texts clearly representing problematizations of the for styles of democratic engagement, as conceptualized above. In light of this, the design logic that underpins the choice of methods is to retrieve a large sample of studies from which there is a good chance to detect at least a few studies that clearly represent each style, if such representations are made in current research. This also means that I do not have the objective to review of all of the retrieved studies. I only analyze those that clearly represent the conceptualized styles of democratic engagement.

To retrieve as many potentially relevant studies as possible in the sample and exclude those not relevant, I first delimited the research to studies on justice issues related to climate change adaptation and resilience policy-making and planning at the local level in developed countries. I retrieved this sample using a Boolean search string (The following Boolean string was used: (TITLE-ABS-KEY ("climate justice" OR "environmental justice" OR "ecological justice" OR "socio-ecological justice" OR "just sustainability")) AND (TITLE-ABS-KEY (adapt* OR resilien* OR vulnerab* OR robust* OR "risk*")) AND (TITLE-ABS-KEY ("climat* chang*" OR "global environmental change" OR "global warming"))). This string was constructed based on keywords that I identified from reading other texts on climate resilience, climate change adaptation and climate justice. The additional keywords in the string are closely related to these three terms. Once constructed, the Boolean string was used to conduct a search in Scopus, on 13 March 2019, where it yielded 340 hits.

The next step used to exclude irrelevant studies was to read the abstracts or the closest equivalent to abstracts of all 340 texts and discard those of low relevance based on the following exclusion criteria:

- Studies not including a focus on countries in the Global North were excluded;
- Studies without a focus on climate resilience, climate change adaptation, or disaster risk reduction were excluded. Studies on disaster risk reduction were included due to their overlaps with climate resilience research;
- Studies not including a focus on the local, urban, or community level were excluded (see introduction);
- Studies in other languages than English and Scandinavian languages were excluded, due to my limited language skills;
- Studies not being peer reviewed articles, books, and book chapters were excluded.

After having used these criteria, 88 studies remained. I read these in their entirety. The key content of the 88 studies was documented in a template designed for literature reviews, based on Cornin, et al. [74]. To this template, I added one row to document content reflecting the political strategies and one to document content reflecting the strategies for social change. This documentation strategy was chosen in order to provide easy access to the core content of these studies, both concerning the strategies they reflect and other important information of these studies, such as the context in which they were set and the methods employed. The latter information was documented to support a richer contextual description of the studies representing any of the four styles of democratic engagement.

Finally, I excluded all studies not representing the problematizations and assumptions reflecting those of one of the groups of political strategies as well as one of the groups of social change strategies. That is, all studies not clearly representing one of the four styles of democratic engagement were excluded. As a result, the number of studies were reduced to ten. Three of them represent problematizations combining the closure- and behavior-oriented strategies, four of them represent problematizations combining the closure- and practice-oriented strategies, and three of them represent a problematization combining the disruptive and practice-oriented strategies. None of the examined studies represent a combination of disruptive and behavior-oriented strategies. Finally, I read these ten studies in their entirety once more, while I wrote the analysis, to confirm that my interpretations of them were plausible.

## 5. Results

This section details the problematizations in the ten analyzed studies that represent different styles of democratic engagement. Specifically, problem representations of the following three styles of democratic engagement were identified: (1) closure-oriented engagement centered on behaviors, (2) closure-oriented engagement centered on practices, and (3) disruptive engagement centered on practices. These problematizations are listed in Table 2.

**Table 2.** Problematizations of Styles of Democratic Engagement.

| | | Political Strategies | |
| --- | --- | --- | --- |
| | | **Closure-Oriented Strategies** | **Disruptive Strategies** |
| **Strategies for social change** | **Behavior-oriented strategies** | Closure-oriented engagement centered on behaviors:<br><br>(1) Low citizen-support for justice-oriented policies is due to a lack of ethical frames and moral reasoning<br>(2) A lack of support for disadvantaged groups to participate in policy and planning processes for adaptation prevents climate justice | Disruptive engagement centered on behaviors:<br><br>- - |
| | **Practice-oriented strategies** | Closure-oriented engagement centered on practices:<br><br>(1) Lack of citizen engagement in deliberations prevents change of systems producing unjust practices<br>(2) Insufficient inclusion of disadvantaged groups prevents change of systems producing unjust practices | Disruptive engagement centered on practices:<br>Insufficient mobilization of an "us" against "them" prevents change of systems producing unjust practices |

### 5.1. Representations of Closure-Oriented Engagement Centered on Behaviors

Problem representations that form part of this style of democratic engagement are implied by prescriptions to promote engagement oriented towards closure on justice-oriented behaviors. As such, they reflect assumptions underpinning both ethically-based democratic theories and ABC theories. Two such problem representations were identified in the research literature. One centers on enhancing the general citizenry's choice to engage with and support justice-oriented policies. The other focuses on ways to encourage and support disadvantaged people to become more engaged in policy and planning processes.

#### 5.1.1. Low Citizen-Support Due to a Lack of Ethical Frames and Moral Reasoning

One problematization of the continuation of "apolitical" approaches is represented in studies that propose ways to enhance the general citizenry's support for justice-oriented climate change adaptation by engaging them through ethical frames and moral reasoning. These proposals imply that the lack of ethical frames and moral reasoning are a problem that results in low public support for justice-oriented adaptation policies. This is based on the assumption that such frames and reasoning tend to activate citizens' ethical motivation to choose to support justice-oriented adaptation—an assumption reflecting the ABC. This problematization is also underpinned by the assumption that politics should be grounded on ethics, as in deliberative and civic republican democratic theories.

Two studies represent this problematization and underlying assumptions. One of these is a study by Adger, et al. [75]. Based on their findings from deliberative discussions between people of different ideological convictions in the United Kingdom, they suggest

"[ ... ] *that there is scope to engage people with climate change adaptation by mobilizing diverse forms of moral reasoning and frames*" ([75], p. 387). They, moreover, emphasize that "[ ... ] *the data also underscores how risks are frequently, and often dominantly, framed in public talk as moral issues rather than issues of economic rationality, likelihood, or individual concern*" ([75], p. 384). Hence, mirroring theories of democracy grounding politics on ethics rather than self-interests, they propose that citizens tend to be concerned with moral issues.

Additionally, these scholars stress that political engagement with moral issues activates citizens' support for policies oriented toward reducing injustices. Reflecting the ABC, ethical frames and moral reasoning is thus presumed to function as a driver motivating support for justice-oriented adaptation policies:

> *The results presented here have important implications for the governance of adaptation, not least in the political legitimacy of different strategies. Despite the diversity of moral reasoning, vulnerability-based motivations, as we define them here, have high salience and are prevalent in public discourse. This suggests that the public is more likely to give support for policies that invest in marginal areas, even at higher cost. Similarly, such motivations are more likely to lend legitimacy to a focus on vulnerability as the key parameter for prioritizing action. This is relevant both for local-level decisions and investments, as it is globally.* ([75], p. 386f)

Although different frames and moral reasoning appeal to different groups of people due to, for instance, diverse ideological convictions, they stress that such reasoning provides avenues to promote discussions of injustices and climate change adaptation focused on offsetting these injustices.

The other study that represents this problematization is by Swim and Bloodhart [76]. This study is also underpinned by the notion that ethical framing and moral reasoning is crucial for promoting democratic engagement with climate injustices. Reviewing (social)-psychology research based on assumptions corresponding to the ABC, these scholars identify cognitive barriers for perceiving climate change as a problem of injustice, especially among privileged groups, and point to ways of surmounting these barriers.

One barrier they discuss is the science and business frame(s) of climate change. As they argue, it is "[ ... ] *important to consider how climate change is publicly discussed* [ ... ]" ([76], p. 481), since "[ ... ] *talking about science and business facilitates technological solutions that maintain superior economic status among industrialized countries, while talking about ethics and justice is more likely to challenge the status quo*" ([76], p. 481). Reflecting civic republican and deliberative democratic theories, they also underscore that encounters with others, particularly vulnerable populations, can promote concerns for injustices and a willingness to decrease these. They, moreover, discuss this in terms of nurturing so-called self-transcendence values, oriented toward the well-being of others and the environment. Mirroring the ABC, these values are presumed to trigger citizens' propensity to choose to support climate-just changes, both through individual and collective actions [76].

### 5.1.2. Lack of Support for Disadvantaged Groups to Participate in Adaptation Processes

Another problematization that forms part of this style of democratic engagement is implied in a study by Kuhl, et al. [77]. This study includes proposals to support disadvantaged populations' participation in policy and planning processes on adaptation. Namely, based on a review of research on evacuation planning for disadvantaged populations in coastal regions in the United States, they offer recommendations to planners that reflect a combination of the assumptions of the ABC and deliberative democratic theories.

One of the recommendations is to "[in]*form and engage EJ* [environmental justice] *communities in decisions about adaptation*" ([77], p. 498). Reflecting the ABC, an assumption of this recommendation is that "[ ... ] *if residents are aware of the long-term plans for their community, they can make informed personal decisions, including the decision to move to other, more protected or less flood-prone neighborhoods*" ([77], p. 498f). In the language of ABC, awareness raising is assumed to be a driver that enables disadvantaged individuals to choose evacuation or to abandon their homes.

Moreover, reflecting deliberative theories' ethical approach to political processes, they propose institutional designs favoring EJ communities' effective engagement in the planning processes: "[ . . . ] *planners and policymakers need to begin a participatory process early, and present community members with sufficient information to effectively engage in planning*" ([77], p. 499).

The emphases on promoting inclusive processes and sufficient information reflect deliberative democratic theories' aspiration to facilitate inclusive communicative processes that enable the participants to reach a rationally based agreement. In this case, on the issues of evacuation and relocation. That is, if individuals in disadvantaged communities are provided access to the planning process at an early stage and are provided sufficient information, it is presumed that their concerns will be promoted. This reflects the notion of deliberative democratic theories, that properly facilitated processes can promote closure based on rational arguments rather than powerful interests.

### 5.2. Representations of Closure-Oriented Engagement Centered on Practices

There are two problematizations that form part of this style of democratic engagement. The assumptions underpinning these reflect a combination of ethically based democratic theories and practice theories. Namely, they set the focus on ways to facilitate ethically grounded political processes to reach agreements of how needs, desires and other attachments to unjust practices are systemically (re)produced and can be unmade and replaced. One of these is centered on promoting the general citizenry's democratic engagement, while the other directs the attention to ways of engaging disadvantaged populations.

### 5.2.1. Lack of Citizen Engagement in Deliberations Prevents Systemic Change of Practices

One problematization of this style of engagement is that a lack of citizen engagement in adaptation planning prevents transformation of the modes of organization that reproduce unsustainable practices and unjust climate vulnerabilities. This problematization is based on assumptions that reflect deliberative democracy and social practice theories.

This problematization is represented in a study by Schlosberg et al. [9], which focuses on local adaptation planning in Australia. These scholars found that the local adaptation plans frame climate change adaptation as a technical problem and stress that this tends to exclude a focus on injustices and silence calls for a systemic transformation of society. However, they also found that a randomly selected panel of 23 citizens from the Sidney area framed adaptation in ways that set the focus on climate injustices and systemic transformation. The focus on injustices encompassed both those in the city and other locations:

> *The citizens immediately expanded the frame from risk alone to risk plus vulnerability, and noted concerns about how risk impacts different populations in very different ways. One of the central demands of the panel was for the City to conduct a broad and thorough vulnerability analysis of these various impacts, in order to gain a better sense of who, exactly, are more likely to be threatened by the risks and impacts of climate change in Sydney. Beyond that, the panel extended their concern for the most vulnerable to those in other, less affluent cities, and asked the City of Sydney to use its resources to assist adaptation planning in the broader Asia Pacific region.* ([9], p. 427)

Hence, the citizens framed adaptation and raised issues that differed markedly from those addressed in the local adaptation plans, particularly expressed through the citizens' ethical concerns for the most vulnerable. This reflects the ethically based democratic theories' assumption that deliberations between citizens promote and nurture a citizen ethos that includes empathy and concern for the wellbeing of others. Moreover, the closure-oriented focus of deliberative democracy emerged through the emphasis that deliberative processes enabled the citizen-participants to reach an agreement on a "*citizen consensus statement*" that encompassed justice-oriented demands, such as those in the passage above ([9], p. 427).

Importantly, the demands that resulted from the deliberative engagement in the citizen panel also encompassed an emphasis on the need for systemic social transformation,

reflecting presumptions underpinning practice theories. The scholars emphasized that citizen engagement could provide avenues to a transformation of unsustainable modes of social and economic organization:

> *If the City of Sydney process is any example,* [ … ] *Citizen engagement* [ … ] *helps address a range of vulnerabilities, different conceptions of potential loss, a broad set of capabilities, and the potential for broad social and economic transformation. It is, in other words and as demonstrated, a necessary component of a process and goal of just adaptation.* ([9], p. 432)

Citizen engagement is thus also assumed to enable "broad social and economic transformation". Related to this, it is furthermore stressed that

> *This transformational language moves away from incrementalism and notions of resilience, representing what some in the literature see as promoting 'the confrontation and questioning of the established systems and their outcomes, [and] tackling the economical, socio-political and cultural roots of vulnerability'.* ([9], p. 432)

It is thus suggested that citizen engagement in adaptation planning can promote systemic critique that mirrors practice theories' emphasis to unmake and replace socio-technical arrangements that reproduce unsustainable practices.

### 5.2.2. Insufficient Inclusion of Disadvantaged Groups Prevents Systemic Change

Another problematization that forms part of this style of democratic engagement is implied in studies that propose ways to facilitate inclusive and ethically grounded policy processes as a way to enable systemic change. It is stressed that these processes should include disadvantaged communities and enable agreement on just adaptation policies that reduce the systemic production of vulnerabilities. The vulnerabilities in focus are those threatening the place-based practices and needs of disadvantaged communities.

One study that represents this problematization is Graham et al.'s article on socially disadvantaged citizens living in low-lying coastal communities in Australia [78]. Reflecting practice theories, they used a mixed methods approach to identify how place- and group-based climate vulnerabilities emerge. They did that to apprehend how sea-level rise can be justly adapted to. Their focus entailed detecting how attachments that different groups of citizens have to particular places and practices were produced. In their article, these attachments are discussed in terms of "*lived values*", which "[ … ] *are not general attitudes that people hold but rather are practices lived by people in places*" ([78], p. 334). A key point is that knowledge of lived values makes it possible to create adaptation policies that respond to the particular vulnerabilities of community-members' place-based practices and attachments rather than to experts' preconceived ideas of what is vulnerable:

> *Understanding the diverse lived values within communities provides local governments with a way of evaluating how climate change impacts, such as sea-level rise, as well as adaptation policies—accommodation, protection and retreat—are likely to affect different groups of people within a community. For example, retreat may be a more acceptable option to the self-sufficient middle aged primary residents and the socially-networked circumstantial seachangers if retirement opportunities and social relationships are maintained, respectively.* ([78], p. 341)

Importantly, it is furthermore emphasized that procedurally fair policy processes that reflect deliberative democratic theories' notion of procedural ethics are important in order for the so-called lived values approach to be realized:

> *From the perspective of procedural fairness, a lived values approach draws attention to the local and the opportunities of shared governance. It enables a more socially nuanced guide to the local politics of adaptation, identifying winners and losers from potential adaptation processes, the diverse needs of constituencies in terms of engagement processes and outcomes, and bundles of adaptation responses that may satisfy some—if not all—members of communities.* ([78], p. 342)

It is, moreover, underscored that such procedural fairness entails "[ . . . ] *that all residents are at least represented in adaptation decision-making*" ([78], p. 341) and that the community should be engaged "[ . . . ] *in robust debates about the trade-offs involved or compensation required for various adaptation options* [ . . . ]" [78], p. 342). An implicit assumption underpinning these proposals is, in line with deliberative democratic theories, that policy processes can become procedurally fair by grounding them on ethics that enable the "better arguments" rather than powerful interest to determine decisions on adaptation policies.

Another example of this problematization is from a study by Miller et al. [79], which examines ways to promote adaptation policies in response to vulnerabilities of disadvantaged African American communities in a coastal area in the USA, namely the eastern shore of the Chesapeake Bay in Maryland. As a result of climate change, this area is increasingly prone to flooding from sea-level rise. Based on an analysis of a workshop with representatives of African American communities as well as representatives of state and federal policymakers, environmental groups, and scholars, these researchers lay out policy recommendations mirroring a combination the assumptions of deliberative democracy and practice theories.

First of all, they stress that African American communities are often excluded from policy processes. As a result of this procedural injustice, adaptation planning and policies recurrently fail to address and respond to the threats that sea-level rise poses to the needs and concerns of these communities. Reflecting deliberative democratic theories' notion of facilitating ethically based procedures, it is emphasized:

> *The results of the workshop highlight the significance of procedural injustice for African Americans on the Eastern Shore of the Chesapeake Bay.* [ . . . ] *In the politics of participation, early engagement with the deliberative process is key for the inclusion of diverse voices* [ . . . ]. [ . . . ] *everyone recognized that sustaining the process to address issues of justice and adaptation will require restructuring institutional and procedural models of governance.* ([79], p. 196)

One of the arguments made for procedural fairness is that policymakers and disadvantaged communities can have, and were found to have, very different ideas of what vulnerability is. Mirroring practice theories' focus on how needs and other attachments transpire through particular arrangements, the scholars underscore that procedurally just processes can promote adaptation policies that respond to the disadvantaged communities' vulnerabilities without destroying elements that support their adaptive capacity:

> *Vulnerability for policymakers is measured with flood maps and social indices, while vulnerability for African Americans is waking up to another day of unemployment, flooded roads, subsiding land or, when hurricanes strike, entire flooded communities. What is at stake are both community resources and the adaptive capacity that is rooted in knowledge of their local environment. Relocation would strip African Americans of a social network they depend upon and a familiar environment to which they have been adapting for generations. Even if relocation protects individuals from flood-related harm, it could cause significant harm by disrupting the strong relationships and culture that have long been fostered in these historical African American communities.* ([79], p. 195)

Reflecting practice theories' focus on how attachments to everyday practices emerge and erode through particular arrangements, the authors thus make two important points in this passage. First, the disadvantaged residents and policy makers did not understand vulnerability in the same way. The former group emphasized systemic failures impeding their basic needs to be met—needs such as access to food, water and shelter. Such failures included a shortage of employment opportunities in combination with relatively few measures to reduce risks of climate-induced floods in these communities. Without procedural justice, such systemic failures would likely not be addressed. Second, in the absence of procedural justice, adaptation planning would risk eroding the adaptive capacity that local knowledge and social networks were found to provide in these communities. In the language of practice theories, it is suggested that procedural justice would promote adaptation

policies that create less vulnerable and more just arrangements, and that prevent the erosion of linkages in these environments that support the adaptive capacity of these communities.

A final example is from a study by Wilder, et al. [80]. Using a mixed methods approach for examining inequalities associated with climate change and climate policies in Southwestern USA (i.e., Arizona and New Mexico), they represent this problematization in relation to climate risks in inland states, such as heatwaves and droughts. Reflecting practice theories, these scholars emphasize that adaptation policies should counter the systemic production of climate-related vulnerabilities—vulnerabilities that threaten particular needs in low-income communities:

> *Communities need to establish emergency preparedness and build local resilience to respond to climate change and ensure climate justice. This requires identifying and addressing the various vulnerabilities of populations according to their unique cultural, socio-economic, and environmental context. In the Southwest, important actions to buffer the complex intersections of climate change and social vulnerability include: housing weatherisation, affordable energy, identification of the most vulnerable populations and emergency planning, job security, and community greening and gardening projects to reduce heat island effects and increase local food security.* ([80], p. 1349)

This passage reflects practice theories' notion of unmaking and replacing unjust socio-technical arrangements. Namely, the focus on "complex interactions" between for instance housing weatherization, job security, affordable energy etc., mirrors practice-theories' systemic focus on socio-technical arrangements.

To enable such adaptation, the authors suggest that inclusive participatory processes are crucial: "*Diverse populations should be incorporated into planning for climate risk and preparedness* [ . . . ]. *Efforts need to be made to broaden the participation of underrepresented groups in climate change preparedness*" ([80], p. 1337). Elsewhere, they indicate that such increase in procedural justice is important to enable disadvantaged low-income communities' definitions and experiences of vulnerability to be voiced and responded to:

> *The Southwest climate gap—the connections between poverty and climate that we have outlined here—is an urgent need to be addressed. [ . . . ] In contrast to the traditional "climate stakeholder"—farmers, water managers, and natural resource planners—the poverty–climate link defines a new set of stakeholders, including vulnerable communities and the government agencies and social services providers that serve them. Thinking across this nexus may yield fresh insights into how climate extremes and climate thresholds are defined, embodied, and experienced by vulnerable communities.* ([80], p. 1349f)

This emphasis on inclusion for procedural justice reflects the notion that policy processes should be grounded on ethics in order to favor agreement based on the "better" argument rather than closure dictated by the interests of powerful actors. That is, the assumption is that it will enable closure on policies to change complex systemic interactions that produce the vulnerabilities defined, experienced, and embodied by disadvantaged strata of the population.

### 5.3. Representations of Disruptive Engagement Centered on Practices

This style of democratic engagement is represented by one problematization implied through prescriptions to mobilize an "us" in order to challenge a privileged "them", and the "unjust" discourses and practices that reproduce their privilege. As such, this problematization sets the focus on engagement to disrupt and dislocate the systemic production of climate injustices. This problem representation is underpinned by assumptions that mirror those of agonistic democratic theories and practice theories. Importantly, the studies that represent this problematization tend to emphasize the need for "just sustainabilities", which entail more than climate change adaptation and resilience but encompass them.

Insufficient Mobilization of an "Us" against "Them" Prevents Systemic Change

This problematization is represented in studies proposing ways of challenging and disrupting naturalized discourses by mobilizing alliances of resistance, thereby creating possibilities for democratic engagement towards systemic change. One example of this is from a study by Di Chiro [5] that, through participatory action research, examines how just sustainabilities, which include resilience building, can be promoted through collaboration between academics and disadvantaged, predominantly black, communities in North Philadelphia, USA. Reflecting agonistic theories' assumption that hegemonies limit the possibilities for democratic politics and therefore should be contested, the study emphasizes that the dominant narrative of the Anthropocene forecloses a focus on environmental injustices. Accordingly, the possibilities for democratic engagement that changes the systemic production of unjust practices and privileges is significantly reduced:

> *Declaring that the planet's core environmental problem is humanity, the Anthropocene story too readily conflates the exploitative cultures and extractive economies of the 1% of high-impact, high-extractive, and high-consumptive humans with the entire species. At the same time, it easily ignores the other large-scale story of global crisis: human inequality and the ongoing struggle for basic human rights for billions of people worldwide. Moreover, the Anthropocene story limits the possibilities for gaining critical insights from examples of sustainable lifeways, knowledges, and cultures that are achieved by those people who have been colonized, enslaved, or eradicated in the service of wealth and domination of the Earth.* ([5], p. 534)

The assumption is thus that the narrative of the Anthropocene masks injustices and privilege and excludes stories of how sustainable modes of organization in marginalized societies are enacted—the latter point mirrors practice theories focus on arrangements from which sustainable practices emerge.

In line with agonistic theories, it is also underscored that the Anthropocene narrative needs to be challenged through peaceful democratic contestation enabled by the mobilization of the narratives of a disadvantaged "us". These narratives should be used, it is suggested, to point to the limitations of the Anthropocene narrative, unsettle its hegemony and open up new possibilities for, among other things, modes organization that produce just forms of resilience. The author calls the latter "mechanisms for collaborative survival":

> *I argue that if these diverse stories of resistance, resilience, resurgence, and what philosopher Kyle Powys Whyte (Potawatomi) calls stories of 'collective continuance' (contesting the individualism of 'sustainability') were noticed, and mattered, we may be able to more fruitfully co-create mechanisms for collaborative survival* [ . . . ]. ([5], p. 534)

In other words, contestation and dislocation of the narrative of the Anthropocene are assumed to create possibilities to construct alternative arrangements. Reflecting the critique of the ABC, put forward by practice-theory scholars, the individualism of sustainability should be challenged during these acts of resistance. The latter is further emphasized in the statement that "[t]*he story of sustainability [ . . . ] has been 'hijacked' by global capitalism's appeals to green individualism (personal lifestyle changes) and hyper-consumerism (vote 'green' with your dollar) [ . . . ]*" ([5], p. 526).

At the local level, Di Chiro [5] suggests that such resistance and dislocation of the Anthropocene narrative can be promoted through collaborations between academics and disadvantaged populations that share an aspiration to mobilize and enact counter-hegemonic narratives and practices. Importantly, these collaborative forms of resistance should be based on the community's embodied experiences and concerns. Her argument is that the latter enables the construction of arrangements that meet the needs of the disadvantaged community and resist arrangements that produce unjust practices of privilege and harm, such as gentrification and displacement. In her study on North Philadelphia, the Sustainable Serenity Collaborative exemplifies this form of collaboration:

> *The Sustainable Serenity Collaborative, now in its sixth year, embraces a vision of sustainability that imagines the possibilities of collaborative survival: it resists and contests*

*green gentrification, supports the values of just sustainability, and is driven by the needs and dreams of the predominantly Black, low-income residents of North Philadelphia whose goals are to develop locally owned, culturally relevant, environmentally conscious, and profitable enterprises that would enable the survival of African American histories and life-ways, and create flourishing refuges [amidst capitalism's ruins] in this corner of the city.* ([5], p. 536)

In short, Di Chiro [5] suggests that resistance against the dominant narrative and practices of sustainability can be mobilized and enacted through collaborations between academics and disadvantaged communities centered on producing modes of organization that benefit the concerns and needs articulated by the latter.

This problematization is also represented in a study based on an action-research approach by Rice, et al. [81]. These scholars argue that scientific knowledge and technical expertise tend to dominate climate-policy making, which silences "[ . . . ] *democratic debate and argument based in a wider discussion of values, norms, and experiences*" ([81], p. 254). Specifically, they draw on agonistic scholars, such as Mouffe and Rancière, and present an "[ . . . ] *approach* [that] *seeks to enhance people's power to make decisions by destabilizing the dominance of scientific knowledge only to create space for more pluralistic knowledge of the problem, its effects, and possible solutions*" ([81], p. 260). Importantly, it is stressed that this is enabled through knowledge co-production between social-science researchers and disadvantaged communities [81].

One presumption of this study is that collaboration between researchers and community groups will result in counter-hegemonic knowledge that can be employed in agonistic debates to challenge the policy-hegemony of scientific knowledge. It is stressed that such democratic debate is crucial "[ . . . ] *because the individuals and communities most often marginalized through expert-only politics are often more likely to experience negative consequences from socioecological changes*" ([81], p. 255).

Rice et al. [81] hence represent a disruptive strategy centered on producing counter-hegemonic knowledge to enable contestation of the techno-scientific hegemony and thereby create possibilities to shift the focus to value-oriented debates on how society should be organized in light of climate change. Particularly, the counter-hegemonic knowledge they seek to mobilize is based on the narratives of marginalized people's embodied experiences, including their experiences of unjust and vulnerability producing social processes:

*These narratives show that residents are keenly aware of their region's connections to the outside, yet they emphasize the regional landscape changes and development processes that contribute to climate change and its impacts far more than distant and global processes. Disentangling climate change from social processes of exurbanization is not possible (or productive) for these individuals. [ . . . ] residents of the region have learned that economic and demographic shifts reshape local landscapes, often to the benefit of wealthy outsiders. The most significant changes today arise from what has been called "amenity migrants" [ . . . ], whose mark on the landscape has made development and its ecological impacts a well-known and controversial issue. One of the most prominent conversations about the interactions between exurbanization and climate change came during highly contested public debates about how steep-slope development—a relatively new form of development driven primarily by amenity migrants—is made riskier by the region's heavy rainfall events.* ([81], p. 258)

As this quote illustrates, it is highlighted that these people's narratives of their embodied experiences can set the focus on debates over social processes from which unsustainable practices emerge. Here it is exemplified with potentially maladaptive practices of exurbanization. This focus on how practices are produced through systemic processes mirrors practice theories' presumption that unsustainable practices are reproduced through socio-technical arrangements and that these need to be unmade and replaced. Accordingly, this study represents a problematization underpinned by the assumption that the narratives of people's embodied experiences and concerns can dislocate techno-scientific hegemony

and promote pluralistic democratic debates on what a just and sustainable society entails, both in light of climate change and the modes of organization that support and undermine sustainability and resilience.

The last study that represents this problematization is an article by Poland et al. [28]. Based on a review of other research, they present ideas of how just, sustainable and resilient environments that support health can be promoted in light of the triple threats of global warming, environmental degradation and peak oil. Reflecting agonistic theories' focus on creating new opportunities through contestation of dominant ways of knowing and doing, they underscore the importance of promoting "[ . . . ] *discontinuous change in profoundly democratic and dialogical ways*" ([28], p. 210). They moreover suggest that there is a need for "[ . . . ] *a paradigm shift from risk management to finding in upheaval the latent possibilities of reconfiguring social and environmental praxis*" ([28], p. 211). This suggestion is combined with a call to challenge and replace systemic modes of organization that reproduce unsustainable and unjust ways of life, as indicative of practice theories. Through alliances with other groups and movements, they argue for

> [ . . . ] *traditional resistance work and organizing aimed at halting or slowing the rate of destruction* [ . . . ]. [ . . . ] *This will require a more explicit alignment in solidarity with indigenous groups and allied social movements, a willingness to value other ways of knowing* [ . . . ]. *This would move the field much closer to critiquing existing social structures that impact environmental health and justice and for sociological sophistication about how contemporary social relations resist an ecological worldview and lifestyle—both of which are preconditions for 'a sense of the possible'*. ([28], p. 209)

The argument is thus that health promotion should be aligned with movements that mobilize other discourses and "ways of knowing" in order to challenge dominant social structures—structures that reproduce unsustainable, unjust and unhealthy practices that undermine disadvantaged groups' resilience.

This resistance movement, furthermore, entails promotion and enactment of alternative arrangements. Referring to practice theory, the authors stress that this encompasses a focus on "[ . . . ] *the conditions under which cultural change on the scale required to realize the vision of 'supportive environments for all' might be catalysed*" ([28], p. 206). For instance, grassroots movements that enact "*communities of practice*" that produce sustainable, resilient and just ways of life ([28], p. 208), and thereby present alternatives to current modes of "*social organization*" ([28], p. 207), should form part of the resistance to dominant modes of organization:

> [ . . . ] *emerging grassroots place-based responses to the triple threat [of global warming, environmental degradation and peak oil] often function as 'communities of practice'—not only shaping local policy, but also helping to create a 'culture of sustainability'*. ([28], p. 208)

In sum, these three studies represent the problem as insufficient mobilization of resistance driven by an "us" that challenges and dislocates currently unjust and harmful modes of organization that benefit a privileged "them". Importantly, a part of this resistance is also to construct and enact alternative arrangements that produce sustainable, just, and resilient practices.

## 6. Discussion

In this section, I discuss the possibilities and limitations produced through the problematizations forming part of each of the three styles of democratic engagement identified in the previous section. This includes how these possibilities and limitations relate to critique launched against the democratic and social change theories that underpin these styles of engagement. I also discuss different ways to approach the different styles, considering the conflicting assumptions on which their problematizations are based.

The closure-oriented engagement on behaviors encompasses problematizations that constitute possibilities to involve the general public as well as empower disadvantaged communities to participate in deliberations on climate injustices. However, one limitation

produced through these problematizations is their silence concerning how unjust and unsustainable practices emerge through particular modes of organization. This silence is constituted by the problematization implied through the prescriptions of Adger et al. [75] and Swim and Bloodhart [76] that focus on ways to activate citizens' ethical and moral inclinations in order to change their policy preferences in favor of justice-oriented adaptation. It is also produced through the problematization in Kuhl et al. [77], which centers on empowering and supporting individuals in disadvantaged communities to participate in policy and planning processes for adaptation. That is, these problematizations constitute individual changes of behavior, and ways to enable and motivate these changes, as the engines of change toward democratic engagement. The lack of democratic engagement thus becomes a matter of choice, either the general citizenry's choice to support vulnerability-reducing policies for disadvantaged strata in the population or marginalized individuals' choice to participate in policy and planning processes.

This silence reflects the critique put forth against the ABC by practice-theory scholars, such as Shove [44]. As she puts it, these theories are "[ . . . ] *a template for intervention which locates citizens as consumers and decision makers and which positions governments and other institutions as enablers*" ([44], p. 1280). This focus on enabling individuals to choose alternative behaviors entails that it is not the systemic modes of organization from which unsustainable practices emerge that are problematized. Rather, the problem is individuals' inaction to choose alternative courses of action. This difference may seem subtle. However, it is arguably important since the latter entails that the continuation of unjust adaptation policies ultimately becomes attributed to individual inaction, given that policy measures have been taken to motivate and enable the individuals to choose sustainable courses of action.

The behavior-oriented style of democratic engagement could of course, on its own, be viewed as one legitimate style of problematizing the lack of democratic engagement. However, considering the open-ended inclination of democratic processes, another approach to this style is to consider it, and the possibilities its problematizations produce, as complementary to the other styles. Namely, it could be viewed as relevant in phases and situations where relatively few people from the general citizenry and disadvantaged groups are engaged with the issues of (un)just adaptation and resilience. There is, in principle, nothing that prevents problematizations that form part of the other styles of democratic engagement to be made once a more widespread democratic engagement is set in motion. If the latter approach is employed, recognition of the limitations of this style of democratic engagement could create possibilities to overcome them through other styles, once people engage with the issue. In this way, the different and conflicting problematizations and assumptions of the other styles could broaden and deepen the engagement further.

The problematizations of the other styles of democratic engagement are, in fact, represented in ways that could support such an eclectic approach. For instance, two possibilities of problematizing social change differently, once democratic engagement is set in motion, come about through the problematizations that form part of closure-oriented engagement on practices.

Regarding the general citizenry, Schlosberg et al. [9] do, as illustrated, represent the lack of citizen engagement in deliberations on climate change adaptation as a problem in the sense that it hampers citizens' calls for systemic change. In places where the citizenry is in agreement that the systemic production of vulnerabilities need to be addressed and changed, as in their study, policies for such change could thus be a substantive outcome that follows enactments of the problematization represented in Adger et al. [75] and Swim and Bloodhart [76].

Concerning disadvantaged communities' engagement, the problematization that insufficient inclusion of them prevents change of systems that produce unjust practices, represented in Graham et al. [78], Miller Hesed and Ostergren [79] and Wilder et al. [80], constitutes a focus on procedural justice. This procedural focus partially reflects that which is constituted through the problematization in Kuhl et al. [77]. Miller Hesed and

Ostergren [79], for instance, also emphasize the need to restructure policy and planning processes to enable earlier participation. This shared emphasis is not surprising considering these problematizations shared grounding on assumptions reflecting those underpinning deliberative democratic theories. The difference is that the problematization represented by this practice-oriented style of democratic engagement constitutes a focus to address and counteract the production of vulnerabilities of disadvantaged groups' place-based practices and needs. This style of democratic engagement thus centers on addressing and responding to emergent vulnerabilities, as these are understood from the perspectives and embodied experiences of disadvantaged groups.

The focus on the systemic production of vulnerabilities—which, it should be recalled, is also constituted through the disruptive style of engagement—could nevertheless be criticized for downplaying individuals' capacities and responsibilities to choose options that reduce vulnerabilities (cf. [43]), especially in comparison to the orientation of the ABC theories. Hence, this is a silence produced through this style of democratic engagement (as well as the disruptive style). Those in favor of attributing most of the responsibilities to individuals would thus not be as likely to participate in the practice-oriented styles of engagement. Knowledge of each styles' possibilities and limitations could thus lead some groups to enact more than one style while others may be less inclined to do so.

With the above in mind, there is also another limitation constituted through the problematizations of both closure-oriented styles of engagement. This limitation follows their shared assumption that policy outcomes can be determined through an ethical foundation (procedural or substantive), rather than the strategic interests of powerful actors. As stressed in the analysis, even the problematizations of systemically produced vulnerabilities represented in Schlosberg et al. [9] as well as in Graham et al. [78], Miller Hesed and Ostergren [79] and Wilder et al. [80] constitute citizens as (potentially) ethical subjects and/or policy and planning processes as (potentially) determined by procedural ethics. They thus either reflect the emphasis on civic virtues in civic republicanism [38,55] or procedural ethics in deliberative democracy [56,57]. To ground politics on ethics in these ways is deemed naïve from the perspectives of, for instance, agonistic [39] and public choice theories [59] of democracy. Hence, although Adger et al. [75] and Schlosberg et al. [9] present empirical evidence in support of this assumption, and Swim and Bloodhart [76] propose ways to develop and cultivate self-transcendence values in support of a citizen ethics, there are reasons to not be too quick to deem this assumption to be generally valid. Notably, Schlosberg et al. [9] suggest a need for such caution. An additional limitation of the problematizations of both closure-oriented styles of engagement is thus that they would be quite useless in places where this assumption is invalid.

The problematization that forms part of the disruptive engagement on practices does, however, constitute possibilities to mobilize resistance against powerful actors and the hegemonic discourses and practices that support their positions and practices. That is, reflecting agonistic theories of democracy, it positions subjects as political adversaries in which an "us" is articulated against a "them". As with agonistic theories, it moreover sets the focus on mobilizing peaceful resistance against "them" and the hegemonic discourses that sustain the order that benefits "them" and not "us". Di Chiro ([5], p. 534), for example, proposes that the environmental crisis is primarily the fault of a privileged 1% of the population. This "1%" is thus constructed as a "them" and in this study most clearly represented by those that benefit from gentrification. Moreover, the Anthropocene is portrayed as a narrative that supports the continuation of the unjust and unsustainable practices enacted by "them". Furthermore, researchers are positioned as subjects that can contribute to mobilize an "us" among disadvantaged communities. Together, researchers and disadvantaged communities should thus enact peaceful resistance that challenges the Anthropocene narrative and the "them" who benefits from it. Put simply, this problematization centers on enacting resistance against the powerful rather than facilitating ethically based agreements between the disadvantaged and the privileged.

The focus on promoting agonistic conflicts between an "us" and "them" has, however, been criticized for neglecting the need for policy closure. Dryzek [82], for instance, argues that agonistic pluralism only focuses on collective decision-making in terms of how collective decisions can be opened to contestation. In one sense, this critique is valid for the problematization represented in Di Chiro [5], Rice et al. [81] and Poland et al. [28]. Namely, it focuses on mobilizing resistance against a "them" and the dominant discourses legitimizing and supporting the continuation of "their" unjust and unsustainable practices. Thereby, there is a silence on how to reach closure for just societal transformations also supported by "them", which reflects the assumption of the "political" in agonistic pluralism [65]. However, in another sense, attention to closure is not absent in this problematization. Namely, it sets the focus on how "we" can come together and mobilize shared narratives and practices that resist "them". In a way, these collective enactments of practices and narratives of resistance represent collective decisions to create something "new" while also contesting dominant discourses and practices. Hence, the silence with respect to closure primarily concerns closure in the sense of agreements between "us" and "them".

As I have implied, the limitation that follows this silence can be considered unavoidable, for instance based on the assumption of "the political" that underpins agonistic pluralism [65]. In that sense, this style of democratic engagement is incompatible with the other styles. However, if this assumption is not deemed unquestionable, the limitation created through this silence could, in a sense, be overcome through closure-oriented styles of democratic engagement. Namely, in some settings, as in that depicted by Schlosberg et al. [9], agreement between "us" and "them" on reducing climate injustices might be reached.

With an eclectic approach, the limitation produced through the ethical assumption of the closure-oriented styles could also be counteracted by employing the disruptive style. For instance, in a scenario where a representative sample of citizens are in agreement on the need for systemic change, as in Schlosberg et al. [9] but fail to gain influential political actors' support for their policy proposals, the situation could be problematized based on the disruptive style of democratic engagement. In this way, the focus would shift towards mobilizing resistance that eventually could challenge and replace unjust systems with those deemed just. Although the assumptions of the different styles of engagement are conflicting and incompatible, I thus argue that it is possible to view the disruptive and closure-oriented styles of engagement as complementary, if an eclectic and pragmatic approach is used.

## 7. Conclusions

This study has conceptualized four styles of democratic engagement and identified problematizations in studies on just climate change adaptation and resilience that represent three of them. It has further examined the possibilities and limitations of the problematizations that form part of each of these three styles. In concluding this paper, the implications of the findings are highlighted, both for researchers in fields such as political ecology and environmental justice, and for policy-makers, planners, activists and other actors outside of academia.

First of all, the representations of the identified styles of democratic engagement offer concrete suggestions of quite different ways to promote democratic engagement for just climate resilience and adaptation at the local level. The degree to which these suggestions are applicable in local contexts other than those from which they are drawn is of course, to varying degrees, uncertain. Nevertheless, the problematizations they represent signify different "conducts of conduct" for promoting democratic engagement. These, and the concrete examples representing them, could function as both an overview of different ways to promote democratic engagement and as points of departure for researchers and other actors—such as policy-makers, planners and activists—interested in advancing a democratization of local climate resilience and adaptation. Although the findings are drawn from studies of local contexts in developed countries, it should also be emphasized that their relevance to local contexts in developing countries cannot be ruled out. The latter could thus be further explored.

Second, since no problematization of the style of disruptive engagement centered on behaviors was represented in the analyzed texts (see Table 2), this raises questions as to why that is and what this style would entail in practice. In some cases, "empty boxes" in matrices can be explained by the circumstance that their constituting parts are incompatible and that they thus represent mere theoretical constructs with no real-life application. However, this explanation is not feasible here. For instance, a common way to mobilize an "us" against "them" that reflects the ABC is to cultivate attitudinal shifts by triggering emotions and raising awareness. Hence, problematizations that represent this style of engagement are indeed possible. As this example illustrates, this style of democratic engagement could for instance set the focus on ways to employ ABC theories to incite a formation of an "us", which is central to disruptive engagement strategies. In light of the political polarization occurring in different parts of the world and the perils it creates [83–85], future research that takes the position of this style could for example explore the possibilities to incite an "us" for local climate justice in ways that both offset hostile relations and promote peaceful contestation with "them"—both of which are core concerns of agonistic pluralism [65].

Finally, by discussing the possibilities and limitations of each style of democratic engagement from the angles of different theoretical perspectives, this study provides insights that enable a reflective and critical approach to each. Whether these styles of engagement are viewed as incompatible or complementary, the findings also point to the importance of "zooming out" and critically view each style from different perspectives. Such multiple-perspective view of the possibilities and limitations of each style is not only important for researchers, but could also be crucial to promote more informed and reflexive political interventions for climate justice, whether these are carried out by policy-makers, planners, activists or other actors. Taken together, this study thus offers a set of important insights that can be used by both researchers and actors outside of academia that are interested in ways in which local climate adaptation and resilience building can be democratized.

**Funding:** This research was funded by the Swedish Civil Contingencies Agency through the project Societal Resilience in Sweden, award number 217-34.

**Institutional Review Board Statement:** Not applicable.

**Informed Consent Statement:** Not applicable.

**Conflicts of Interest:** The author declares no conflict of interest. The funders had no role in the design of the study; in the collection, analyses, or interpretation of data; in the writing of the manuscript, or in the decision to publish the results.

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
