# Peer review of "From Technocracy to Democracy: Ways to Promote Democratic Engagement for Just Climate Change Adaptation and Resilience Building"

_sustainability, doi:10.3390/su14031433_

Round 1

Reviewer 1 Report

This study discusses the conflicts between democratic engagement in climate injustice and the dominance of technocrats in this issue based on the contexts of developed countries.

  1. This research is based on a thorough and systematic literature review but how this review could be translated to the current findings is not clear enough. More explanation is required.
  2. The discussions of the study are only based on the contexts of developed countries. If the current discussions of the study are applied to the contexts of less developed or developing countries, what should be considered needs to be discussed too.
  3. How the current findings could give “practical” lessons and guidance to real policymakers, activists, and practitioners should be more discussed.

Author Response

First of all I'd like to thank the reviewer for reading my paper and for providing comments on how it can be improved. Below I've listed the reviewers comments and my responses to them. 

Comment 1: This research is based on a thorough and systematic literature review but how this review could be translated to the current findings is not clear enough. More explanation is required.

Answer: Thank you for this comment. As I see it, it is based on a misunderstanding of my research design, which calls for a clarification of my intent (both here and in the paper). The point with initially retrieving a large sample of studies (using methods also employed for systematic literature reviews) was to increase the chance to detect studies that clearly exemplify the styles of democratic engagement I conceptualize in my paper. Once detected, these would then provide illustrations of what the styles of engagement could entail in practice related to climate (in)justice in local climate adaptation and resilience building. Put simply, the logic of the design is that a large sample of studies, ceteris paribus, increases the likelihood of finding good illustrations of the four styles of democratic engagement. My ambition was never to make a review of all the studies or to translate/relate the content of the studies that form part of the whole sample to the findings of the few studies that exemplify the styles of democratic engagement. The findings presented in the results section are instead discussed in relation to literature on democratic theories and theories of social change. I can see how my design caused confusion and have made an effort to clarify the intent and logic of my design and use of the methods for data retrieval in the methods section.

Comment 2: The discussions of the study are only based on the contexts of developed countries. If the current discussions of the study are applied to the contexts of less developed or developing countries, what should be considered needs to be discussed too.

Answer: This is an important question. Although it is difficult to provide an adequate answer to it, considering that developing countries were excluded from my sample, I think that my paper benefits from highlighting that the findings are potentially relevant for local adaptation and resilience in developing countries as well, and that this can be further explored. I should also add that the question you raise here made it clear to me that I had not adequately elaborated on the implications of my findings. After all, the relevance of the concrete examples of ways to promote democracy that are analyzed in my study is, to varying degrees, uncertain beyond the contexts from which they are drawn. In response to this comment, I therefore developed the concluding elaboration of the implications of my findings for researchers and other actors interested in ways that local climate change adaptation and resilience building can be democratized.

Comment 3: How the current findings could give “practical” lessons and guidance to real policymakers, activists, and practitioners should be more discussed.

Answer: This comment is made by the other reviewer as well. Clearly, this part of the paper needs clarification. What I mean to say is that, on the one hand, my study provides examples of how different styles of democratic engagement or “conduct of conduct” can be promoted in practice, as illustrated in my results chapter. For instance, employment of ethical frames and moral reasoning have been shown to promote citizens’ interest in and support for justice-oriented policies. Likewise, facilitation of citizen engagement in deliberative processes have been found to promote a shift towards discussing and responding to climate injustices (these are only two of several examples provided in the results section). On the other hand, my study provides a discussion of the possibilities and limitations. This discussion offers insights for a more reflective approach to the different ways in which democratic engagement can be promoted. In short, I argue that I provide an overview and analysis with concrete examples of different ways in which democratic engagement for climate justice can be promoted and a discussion that can inform a reflective approach to these. I have clarified this argument in the conclusions where I, as mentioned, have developed my discussion of the implications of my findings, including the implications in terms of “practical” lessons.    

Reviewer 2 Report

Review manuscript sustainability 1520274

From Technocracy to Democracy: Ways to Promote Democratic 2 Engagement with Climate (In)Justice Related to Adaptation 3 and Resilience at the Local Level

This manuscript presents the results of a systematic literature review executed to explore the value and applicability of a deliberately developed framework to evaluate four styles of democratic engagement. Focus is on the way democratic engagement is represented and possibilities and limitations of each style. Theory and methodology are fine. It is very decent work.

In Ch7 Conclusions(923-927) the author(s) write: “knowledge of these styles of democratic engagement, and the possibilities and limitations they constitute, is also of particular relevance to policy-makers, planners, citizens/residents interested in how democratic engagement with cli-mate (in)justice could be advanced. After all, it is arguably this study’s potential to incite ideas and actions among actors outside of academia that is most important for a shift from technocracy to democracy.” This is where my doubts are explained by the author(s) themselves. The article is heavy in theory and conceptualizations and I had a hard job to stay concentrated and follow the argumentation. It is sound science, but hard work as it is all theory and hardly rooted in practice. Every time a little piece of empirical proof was presented it suddenly got interesting! As a practitioner I have been wondering from the very start about the relevance for practice and now reaching the end I find myself disappointed that the best I get is the suggestion not to get caught in one approach but to make a nice mix? Well, I could have told you that before; don’t need a difficult piece of theoretical analysis for that!

For more theory-oriented colleagues I think this might be a nice read though and the subject of environmental justice is timely and well-chosen. But if not I had been the assigned reviewer I would probably have left after a few pages….

Some minor things I would like to bring to attention:

On Discussion:  I had hoped in the 'Discussion' section to find some thoughts on why the quadrant

"Disruptive engagement centred on behaviours" has remained empty? The author(s) does not say anything on this disappointing result of the literature research? I think it is intriguing!  Do the characteristics of the constituting elements rule out any fitting behaviour or might we need to explore how we could tick this box and what it would add to our opportunities? Can the author(s) speculate about it, for as far as I can see it is the only surprising outcome?  

On English language: not being a native speaker myself I think it is perfect. If only in the Discussion section there are (I think) a few typos:

808 production of vulnerabilities

818 emphasize the need to restructure

852 there are reasons to not be too quick to deam the

Author Response

First of all I want to thank the reviewer for taking the time to read and comment on my paper. I really appreciate the comments on how it can be improved. Below, I've listed the comments and my replies to them. 

Comment: In Ch7 Conclusions (923-927) the author(s) write: “knowledge of these styles of democratic engagement, and the possibilities and limitations they constitute, is also of particular relevance to policy-makers, planners, citizens/residents interested in how democratic engagement with climate (in)justice could be advanced. After all, it is arguably this study’s potential to incite ideas and actions among actors outside of academia that is most important for a shift from technocracy to democracy.” This is where my doubts are explained by the author(s) themselves. The article is heavy in theory and conceptualizations and I had a hard job to stay concentrated and follow the argumentation. It is sound science, but hard work as it is all theory and hardly rooted in practice. Every time a little piece of empirical proof was presented it suddenly got interesting! As a practitioner I have been wondering from the very start about the relevance for practice and now reaching the end I find myself disappointed that the best I get is the suggestion not to get caught in one approach but to make a nice mix? Well, I could have told you that before; don’t need a difficult piece of theoretical analysis for that! For more theory-oriented colleagues I think this might be a nice read though and the subject of environmental justice is timely and well-chosen. But if not I had been the assigned reviewer I would probably have left after a few pages….

Answer: Thank you for this valuable comment. If you interpret the only message of my study to be “to make a nice mix”, I clearly need to communicate better what I want to say, regarding the utility for practitioners. To put it briefly, I would argue that my study both provides concrete examples of different ways to promote democratic engagement as well as insights of the possibilities and limitations of each – insights that are important for a more reflective approach. An important point with the theories and theoretical discussion used in the paper is to provide different perspectives on each style of engagement (or “conduct of conduct”), thereby offering tools for the reader to view these from different angles. It that way, the theories enable the reader to “zoom out” and view each style from a “distance”. At times, the discussion is perhaps a bit heavy for some readers, but since my target group is diverse and also include researchers using these theories, those parts of the paper serve a purpose. As mentioned, another key message, in addition to the reflective insights provided by different perspectives, is that I offer an overview and points of departure for several concrete ways to promote democratic engagement, each categorized according to a particular “conduct of conduct” or style of engagement, as reported in the results section. I thus offer concrete examples of different styles of democratic engagement and how they can be promoted in practice. One example of this is to employ ethical frames and moral reasoning to promote citizens’ interest in and support for justice-oriented policies. Another is to facilitate deliberative processes with citizens to promote a shift towards discussions and responses that address climate injustices. In response to your comment, I have clarified these main take-home messages in the conclusions, which is completely rewritten to emphasize the implications of my findings for researchers and practitioners alike.

Comment: Some minor things I would like to bring to attention: On Discussion:  I had hoped in the 'Discussion' section to find some thoughts on why the quadrant "Disruptive engagement centred on behaviours" has remained empty? The author(s) does not say anything on this disappointing result of the literature research? I think it is intriguing!  Do the characteristics of the constituting elements rule out any fitting behaviour or might we need to explore how we could tick this box and what it would add to our opportunities? Can the author(s) speculate about it, for as far as I can see it is the only surprising outcome?

Answer: I think this is an important point. When writing the first draft, I thought about including such a discussion, but choose not to since it would prolong an already long paper. Reading the paper again, I do however agree an elaboration of this would improve the paper. I’ve added this in the conclusions.

Comment: On English language: not being a native speaker myself I think it is perfect. If only in the Discussion section there are (I think) a few typos: 808 production of vulnerabilities, 818 emphasize the need to restructure, 852 there are reasons to not be too quick to deam the

Answer: Thank you for taking the time to check for typos. I’ve looked over these and other parts of the text and corrected the detected errors.

Round 2

Reviewer 1 Report

My comments and suggestions have been incorporated successfully by the author(s).